# Valorization of Linen Processing By-Products for the Development of Injection-Molded Green Composite Pieces of Polylactide with Improved Performance

**Ángel Agüero [1], Diego Lascano [1] , David Garcia-Sanoguera [1], Octavio Fenollar [1] and Sergio Torres-Giner [2],\***

[1]  Technological Institute of Materials (ITM), Universitat Politècnica de València (UPV), Plaza Ferrándiz y Carbonell 1, 03801 Alcoy, Spain; anagrod@epsa.upv.es (Á.A.); dielas@epsa.upv.es (D.L.); dagarsa@dimm.upv.es (D.G.-S.); ocfegi@epsa.upv.es (O.F.)

[2]  Novel Materials and Nanotechnology Group, Institute of Agrochemistry and Food Technology (IATA), Spanish National Research Council (CSIC), Calle Catedrático Agustín Escardino Benlloch 7, 46980 Paterna, Spain

\*   Correspondence: storresginer@iata.csic.es; Tel.: +34-963-900-022

**Abstract:** This work reports the development and characterization of green composites based on polylactide (PLA) containing fillers and additives obtained from by-products or waste-streams from the linen processing industry. Flaxseed flour (FSF) was first produced by the mechanical milling of golden flaxseeds. The resultant FSF particles were melt-compounded at 30 wt% with PLA in a twin-screw extruder. Two multi-functionalized oils derived from linseed, namely epoxidized linseed oil (ELO) and maleinized linseed oil (MLO), were also incorporated during melt mixing at 2.5 and 5 parts per hundred resin (phr) of composite. The melt-compounded pellets were thereafter shaped into pieces by injection molding and characterized. Results showed that the addition of both multi-functionalized linseed oils successfully increased ductility, toughness, and thermal stability of the green composite pieces whereas water diffusion was reduced. The improvement achieved was related to both a plasticizing effect and, more interestingly, an enhancement of the interfacial adhesion between the biopolymer and the lignocellulosic particles by the reactive vegetable oils. The most optimal performance was attained for the MLO-containing green composite pieces, even at the lowest content, which was ascribed to the higher solubility of MLO with the PLA matrix. Therefore, the present study demonstrates the potential use of by-products or waste from flax (*Linum usitatissimum* L.) to obtain renewable raw materials of suitable quality to develop green composites with high performance for market applications such as rigid food packaging and food-contact disposable articles in the frame of the Circular Economy and Bioeconomy.

**Keywords:** PLA; flax; multi-functionalized vegetable oils; green composites; waste valorization

## 1. Introduction

One of the main current objectives of the scientific community is the research and development of novel and high-performance sustainable materials that can replace polymers obtained from petroleum. Green composites represent the next generation of sustainable materials since they combine natural fillers, both particles and fibers, with biopolymer resins to make light and strong articles that are not only fully bio-based but also biodegradable [1]. Green composites also offer environmental advantages in comparison with traditional polymer composites, such as reduced dependence on non-renewable energy/material sources, lower greenhouse gas and pollutant emissions, and improved energy recovery [2]. Furthermore, they show a potential reduction of both product density and energy

requirements for processing [3]. Bio-based industries are currently looking for innovative technologies based on non-edible biomass and industrial by-products and wastes as the main source for bioplastics, which includes large amounts of agro-food residues as well as non-food crops or cellulosic biomass that are abundant and have currently low economic value [4]. Polylactide (PLA) is currently gaining great importance as the matrix in green composites due to its dual advantages of being bio-based and biodegradable [5]. This biopolyester is nowadays considered the front runner in the bioplastics market, with an annual consumption of 140,000 tons [6]. PLA is of particular interest in the manufacturing of composites for various industrial sectors such as rigid packaging and building and construction [7].

The use of lignocellulosic fillers in biopolymers offers significant processing advantages, biodegradability, non-abrasive, low cost, high specific strength, and a renewable nature [8]. Over the last few years, several articles and applications of green composites have been developed using lignocellulosic fillers derived from food, agricultural, industrial, and marine wastes such as rice husk [9,10], almond shell [11], walnut shell [12], peanut shell [13], coconut fibers [14], orange peel [15], recycled cotton [16], and posidonia oceanica seaweed [17]. In this regard, flax (*Linum usitatissimum* L.) is one of the most researched plants for industrial purposes. This plant has been cultivated and used for millennia. Indeed, it belongs to the "founder crops" that are associated with the initiation of agriculture about 8000 B.C. and, thus, it is highly adapted to different environmental conditions throughout all the continents [18]. The five major flax producing countries are Canada, China, India, USA, and Ukraine, jointly accounting for ~72% of the world production [19]. Approximately 230 species of this crop have been reported, the Ukrainian one being the most common. The flax plant varieties are typically distinguished as common flax (for bast fiber production) and linseed (for seed oil production). Flax is mainly cultivated to obtain its linen yarns, whose garments are valued for their exceptional coolness and freshness in hot and humid weather. The flax fibers are characterized by a high modulus and wear resistance compared to other lignocellulosic fibers [20]. Flaxseed has been traditionally a processing by-product of the textile industry, though its industrial use has extensively grown in the last years to extract its oil for the manufacture of paints, varnishes, linoleum, oilcloths, printing inks, and soaps. The interest in linseed oil comes from its exceptionally high content in polyunsaturated fatty acids, mainly α-linolenic acid and lignans, whose positive health effects have been widely demonstrated [21]. Since the whole flax worldwide production was approximately 2.8 million tons from a land area of 2.7 million hectares in 2017 and the oil concentration in the seeds is estimated between 32.4 and 46.4 wt%, it is considered that approximately 1.5–1.8 million tons of waste will be annually discarded [22]. Linseed meal, which is also a by-product of producing linseed oil from its seeds, is currently used as livestock fodder, though the benefits on its consumption are still being investigated. In other cases, it is simply treated as a waste for composting or incinerated [23]. Therefore, the remaining highly lignified cellulosic material obtained after the flaxseed oil extraction has not found yet profitable applications whereas it is a potential candidate to be applied as renewable fillers to reinforce biopolymers.

Vegetable oils can easily find several applications in the chemical and plastic industry, such as plasticizers, paints, resins, or adhesives [24,25]. These oils are mainly composed of triglycerides, produced by the esterification of glycerol with three fatty acids, which can present up to three double bonds per fatty acid. Linseed oil has one of the highest content of polyunsaturated fatty acids [26]. Thus, introducing epoxy, hydroxyl or maleic functionalities into these double bonds can provide reactivity to linseed oil, which results in novel additives that can provide chain-extension, branching, and/or cross-linking reactions for polymer materials [27]. Therefore, multi-functionalized linseed oils are very attractive for new sustainable polymer applications due to its availability, inexpensive cost, and intrinsic biodegradability. For instance, Mosiewicki et al. [28] synthesized a rigid thermoset using linseed oil monoglycerides. Novel vinyl plastisol/wood flour composites with enhanced compatibilization were prepared by plasticizing poly(vinyl chloride) (PVC) with 70 parts per hundred resin (phr) of epoxidized linseed oil (ELO) and then applying ultraviolet (UV) light [29]. In another work, ELO was used as both plasticizer and reactive compatibilizer in green composites made of PLA and hazelnut shell flour (HSF) [30]. Mahendran et al. [31] employed acrylated epoxidized linseed oil (AELO) to obtain

a photo-curable coating applicable on wood surfaces. Linseed oil can also be modified with maleic acid (MA) to develop maleinized linseed oil (MLO), which can successfully improve the ductility and toughness of PLA articles [32].

　　This work aims to develop, for the first time, a green composite based on PLA containing fillers and additives completely derived from linen processing by-products and waste. To this end, flaxseed flour (FSF) was first obtained from flaxseeds by mechanical milling, and then the particles were incorporated by melt compounding into PLA at 30 wt%. During the extrusion process, ELO or MLO were added at 2.5 phr and 5 phr of green composite to improve the interfacial adhesion between the dispersed lignocellulosic particles with the biopolymer matrix. The resultant green composite pellets were finally shaped into pieces by injection molding and characterized in terms of their thermal, mechanical, and thermomechanical properties and also water absorption and diffusion characteristics to ascertain their application in the food packaging industry.

## 2. Materials and Methods

### 2.1. Materials

　　PLA grade Ingeo Biopolymer 6201D was supplied in pellet form by NatureWorks LLC (Minnetonka, MN, USA). This grade is suitable for injection molding since it has a melt flow rate (MFR) of 15–30 g/10 min, measured at 210 °C and 2.16 kg.

　　FSF was obtained from golden flaxseeds, which were provided by NaturGreen (Murcia, Spain). The whole seeds were flat and oval with pointed tips, showing a brown color and measuring approximately 2.5 mm × 5 mm × 1.5 mm. The seeds were subjected to a grinding process in an ultra-centrifugal mill of Retsh GmbH (Hann, Germany) with a rotation speed of 8000 rpm and meshed to produce a mean particle size below 0.25 mm. Figure 1 shows the as-received flaxseeds and the resultant flour.

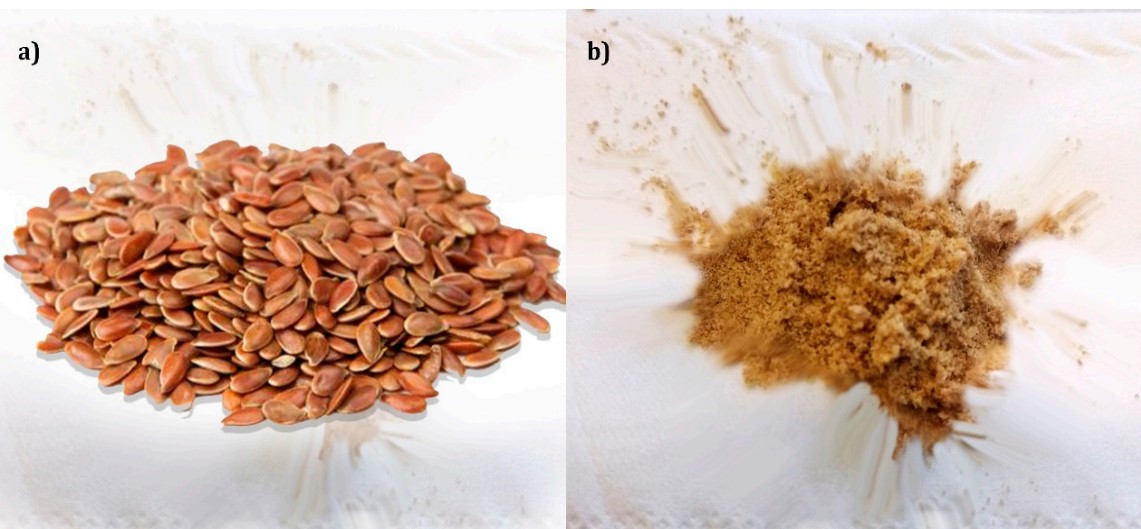

**Figure 1.** (**a**) As-received golden flaxseeds; (**b**) flaxseed flour (FSF).

　　ELO, CAS number 8016-11-3, was supplied by Traquisa S.L. (Barcelona, Spain). It shows a density of 1.06 g/cm$^3$ measured at 20 °C, a molecular weight (M$_W$) of approximately 1037 g/mol, and a viscosity of 8000–11,000 cP at 25 °C. Its oxirane oxygen content (OOC) is nearly 8% and its iodine value (IV) is ≤5%. The average fatty acid profile is 56–51% linolenic acid, 26–18% oleic acid, 20–14% linoleic acid, 7–5% palmitic acid, and 5–3% stearic acid. It shows high thermal stability with a flammability point at 287 °C, being not soluble in water. MLO, CAS number 68309-51-3, was provided by Vandeputte (Mouscron, Belgium) as VEOMER LIN. It presents an acid value of 105–130 mg potassium hydroxide (KOH)/g and a viscosity of approximately 1000 cP at 20 °C.

## 2.2. Preparation of Green Composite Pieces

Prior to extrusion, the PLA pellets and FSF, in powder form, were dried separately for 24 h at 60 °C in a dehumidifying dryer MDEO from Industrial y Comercial Marsé S.L. (Barcelona, Spain) to reduce moisture and avoid hydrolysis. Then, different quantities of the raw materials, that is, PLA, FSF, ELO, and MLO, were weighed and pre-mixed in a zipper bag. The set of formulations prepared are shown in Table 1. The filler-to-matrix ratio was set in all cases at 30 wt% since this content has shown the most optimal results in our previous studies dealing with green composites of PLA [33].

**Table 1.** Summary of compositions according to the weight content (wt%) of polylactide (PLA) and flaxseed flour (FSF) in which epoxidized linseed oil (ELO) and maleinized linseed oil (MLO) were added as parts per hundred resin (phr) of composite.

| Samples | PLA (wt%) | FSF (wt%) | ELO (phr) | MLO (phr) |
|---|---|---|---|---|
| PLA | 100 | 0 | 0 | 0 |
| PLA/FSF | 70 | 30 | 0 | 0 |
| PLA/FSF + 2.5ELO | 70 | 30 | 2.5 | 0 |
| PLA/FSF + 5ELO | 70 | 30 | 5.0 | 0 |
| PLA/FSF + 2.5MLO | 70 | 30 | 0 | 2.5 |
| PLA/FSF + 5MLO | 70 | 30 | 0 | 5.0 |

A twin-screw co-rotating extruder from DUPRA S.L (Castalla, Spain) was employed to compound the different formulations. The temperature profile during extrusion was set, from the hopper to the die, at 165–170–175–180 °C. A rotating speed of 40 rpm was used. Pelletizing of the resultant air-cooled extruded strands was performed in an air-knife unit.

A Meteor 270/75 injection molding machine from Mateu & Solé S.A. (Barcelona, Spain) was used to manufacture the pieces from the melt-compounded pellets. The temperature profile during injection molding was set, from the feeding zone to the injection nozzle, at 180–165–165–65 °C. The cavity filling, packing, and cooling times were set at 1 s, 10 s, and 10 s, respectively. The clamping force during the process was 75 tons. The mean thickness of the pieces was approximately 4 mm.

## 2.3. Characterization of Green Composite Pieces

### 2.3.1. Mechanical Tests

Tensile tests of the green composite pieces were obtained using a universal test machine ELIB 30 from S.A.E. Ibertest (Madrid, Spain), as indicated in the UNE-EN-ISO 527-1 standard. The pieces sized 150 mm × 10 mm × 4 mm and the selected load cell and cross-head speed were set 5 kN and 5 mm/min, respectively. An axial extensometer, model IB/MFQR2, also from S.A.E. Ibertest, was coupled to the tested samples to improve accuracy [34]. Impact strength was tested in a 6-J Charpy pendulum from Metrotec S.A. (San Sebastián, Spain) following the guidelines of ISO 179. Unnotched pieces with dimensions of 80 mm × 10 mm × 4 mm were used. Finally, hardness was measured in a Shore D durometer model 673-D from J. Bot S.A. (Barcelona, Spain), according to ISO 868. At least six different samples were tested for each mechanical property and the values were averaged.

### 2.3.2. Microscopy

Field emission scanning electron microscopy (FESEM) was used to evaluate the fracture surfaces of the green composite pieces obtained from the impact tests. A ZEISS ULTRA 55 FESEM microscope from Oxford Instruments (Abingdon, United Kingdom) was used, working at an acceleration voltage of 2 kV. Samples surfaces were coated, prior to analysis, with a gold-palladium alloy in a Quorum Technologies Ltd. EMITECH model SC7620 sputter coater (East Sussex, UK).

### 2.3.3. Thermal Tests

Differential scanning calorimetry (DSC) was employed to evaluate the thermal transitions of the green composite pieces using a Mettler-Toledo 821 calorimeter (Schwerzenbach, Switzerland). Samples with an average weight of 5–8 mg were placed in a standard 40 μL aluminum sealed crucibles and subjected to a three step program that consisted of an initial heating cycle from 25 °C to 200 °C to remove thermal history, followed by a cooling to 0 °C, and a second heating cycle to 350 °C. All the tests were run at 10 °C/min in an inert atmosphere with a constant flow of nitrogen of 30 mL/min. The crystallinity degree ($X_c$) was calculated using Equation (1):

$$X_C(\%) = \left[\frac{\Delta H_m - \Delta H_{CC}}{\Delta H_m^0 \cdot (1 - w)}\right] \times 100 \tag{1}$$

where $\Delta H_m$ and $\Delta H_{CC}$ (J/g) correspond to the melting and cold crystallization enthalpies of PLA, respectively; $\Delta H_m^0$ (J/g) is the theoretical value of a fully crystalline PLA, that is, 93.0 J/g [35]; and 1-*w* indicates the weight fraction of PLA in the green composites.

Thermogravimetric analysis (TGA) was used to study the thermal stability of the green composite pieces using a TGA/SDTA 851 thermobalance from Mettler-Toledo Inc. A single-step thermal program was selected from 30 °C to 700 °C at a constant heating rate of 20 °C/min in air atmosphere. All the thermal tests were performed in triplicate.

### 2.3.4. Thermomechanical Tests

Dynamic mechanical thermal characterization (DMTA) was carried using an AR-G2 oscillatory rheometer from TA Instruments (New Castle, DE, USA). This device was equipped with a clamp system for solids working in a combination of torsion and shear. Rectangular pieces of 10 mm × 40 mm × 4 mm were subjected to a temperature sweep from 30 °C to 140 °C at a heating rate of 2 °C/min and a constant frequency of 1 Hz. The maximum shear deformation percentage (%γ) was set to 0.1%. All the thermomechanical tests were performed in triplicate.

### 2.3.5. Water Absorption Analysis

Water absorption was carried out with distilled water at 30 ± 1 °C for a period of 130 days, according to the ISO 62:2008 standard. The size of the pieces was 80 mm × 10 mm × 4 mm and, prior to water immersion, the samples were dried at 60 °C for 24 h in an air circulating oven model 2001245 Digiheat-TFT from J.P. Selecta S.A. (Barcelona, Spain) to remove residual moisture. Samples were taken out of the immersion bath once a week, dried with a dry cloth, weighed in an analytic balance model AG245 from Mettler Toledo Inc., and immersed again in the water bath. All the water absorption tests were carried out in triplicate to ensure reliability. The total absorbed water ($\Delta m_t$) during water immersion was calculated following Equation (2):

$$\Delta m_t(\%) = \left(\frac{W_t - W_0}{W_0}\right) \times 100 \tag{2}$$

where $W_t$ is the sample weight after an immersion time *t* and $W_0$ is the initial weight of the dry sample before immersion.

The value of diffusion coefficient (*D*) was estimated by the application of the Fick's first law in the linear region of the weight gain plot, where $W_t/W_s \leq 0.5$, plotting $\Delta m_t = f(\sqrt{t})$, according to Equation (3):

$$\frac{W_t}{W_s} = \frac{4}{d} \cdot \left(\frac{D \cdot t}{\pi}\right)^{1/2} \tag{3}$$

where $d$ represents the initial thickness of the sample and $W_s$ is the saturation weight, that is, the weight obtained when no additional weight gain is observed. A plot representation of $W_t/W_s$ versus $t^{1/2}$ allows an estimation of $D$ using Equation (4):

$$\mathbf{D} = \pi \cdot \left( \frac{\mathbf{d \cdot \theta}}{4} \right)^2 \tag{4}$$

where $\theta$ corresponds to the slope of the plot. Since the above expression is only valid for a one dimensional shape, the Stefan approximation was used. It considers different corrections to make this expression useful for three dimensional shapes, as shown in Equation (5) [36]:

$$\mathbf{D_c} = \mathbf{D} \left( 1 + \frac{\mathbf{d}}{\mathbf{h}} + \frac{\mathbf{d}}{\mathbf{w}} \right)^{-2} \tag{5}$$

where $D_c$ is the corrected diffusion coefficient that is related to the geometry, while $h$ and $w$ represent the total sample length and width, respectively. This equation is based on the assumption that the diffusion rates are the same for all directions.

## 3. Results

### 3.1. Morphology of FSF Particles

Figure 2 shows the morphology of the FSF particles, that is, of the flaxseed particles obtained after grinding. In the general view of the resultant FSF, shown in Figure 2a, one can observe that the particles presented a round and irregular shape. However, in Figure 2b, which corresponds to a FESEM image taken at higher magnification, it can be seen that the particle surface was relative rough, showing high porosity and some granular fractures. These morphological characteristics probably result from the crushing process due to the hardness of this type of lignocellulosic particles. A similar morphology has been recently described for other lignocellulosic particles derived from similar crop wastes [12,37]. The size distribution presented in the particle histogram, plotted in Figure 2c, revealed that the particle sizes varied in the range of 1–35 μm whereas the mean size was ~15 μm.

### 3.2. Visual Aspect of PLA/FSF Composite Pieces

Injection molding represents a cost-competitive melt-processing methodology to produce a large number of green composite pieces and parts. In this process, the PLA resin and the FSF fillers were fed together into the rear part of the injection screw. Once the biopolymer was melted by the reciprocating screw against the unmoving barrel wall, a metered dosage of polymer melt was transferred towards the nozzle of the injection unit and then injected into the mold, where the heat from the melt dissipates rapidly. In general, very high shear rates arise in injection molding operations, usually up to $10^4$ s$^{-1}$ [38]. A cavity filling time of 1 s was set during mold cavity filling to optimize the process, whereas a packing time of 10 s during back pressure was used to improve the final quality and properties of the green composite pieces. After 10 s of cooling, the PLA-based pieces were rigid enough and they were ejected from the mold to complete cycle. Figure 3 shows the visual aspect of the resultant neat PLA piece and the green composite pieces. One can observe that all the injection-molded PLA pieces were uniform with a bright surface and free of defects. The neat PLA piece presented a natural bright color and it was opaque, indicating that the biopolymer developed crystallinity during cooling in the injection mold. The green composite pieces showed an intense brown color, being assigned to the natural color of FSF, as shown in previous Figure 1b. One can also observe that the introduction of the multi-functionalized vegetable oils slightly reduced the brown tonality in the pieces. The wood-like color and surface finish of these green composite pieces can be of interest from an aesthetical perspective for compostable packaging uses such as food trays for fruits and vegetables or disposable food-contact articles such as cutlery and straws [11].

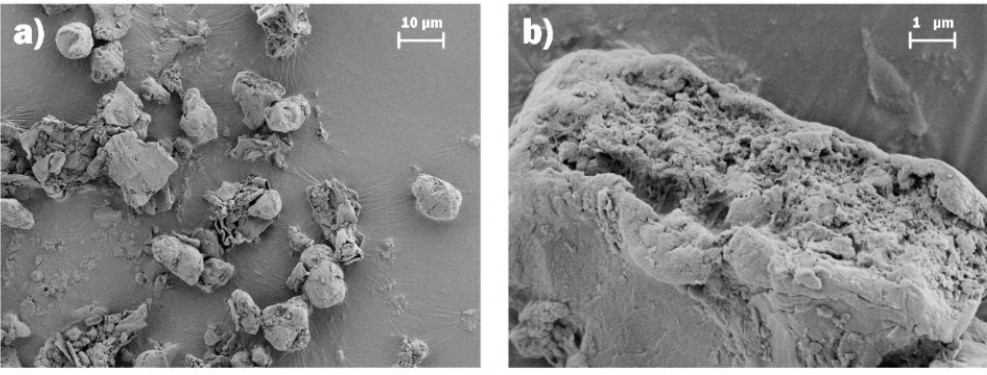

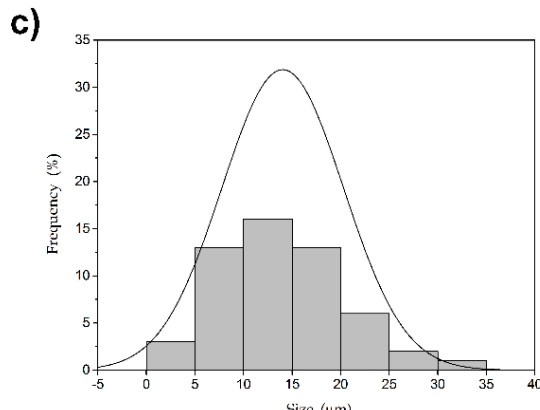

**Figure 2.** Field emission scanning electron microscopy (FESEM) images of flaxseed flour (FSF) taken at (**a**) 1000× with a scale marker of 10 μm and (**b**) 10000× with a scale marker of 1 μm; (**c**) particle size histogram of FSF.

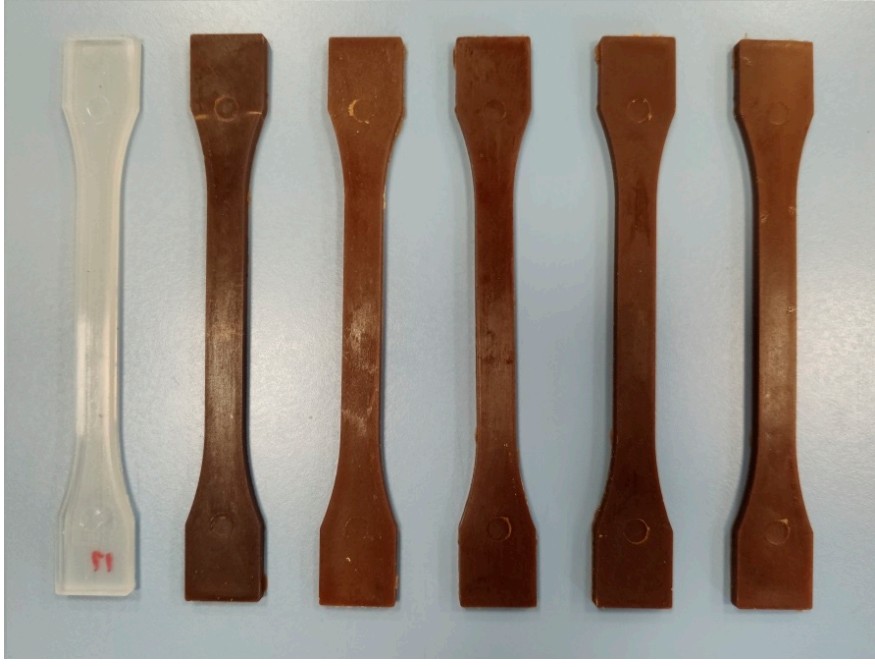

**Figure 3.** Visual aspect of the polylactide (PLA)/flaxseed flour (FSF) composite pieces processed with epoxidized linseed oil (ELO) and maleinized linseed oil (MLO) corresponding, from left to right, to: PLA, PLA/FSF, PLA/FSF + 2.5ELO, PLA/FSF + 5ELO, PLA/FSF + 2.5MLO, and PLA/FSF + 5MLO.

## 3.3. Mechanical Properties of PLA/FSF Composite Pieces

Table 2 gathers the results of the mechanical tests for the injection-molded pieces of neat PLA and the PLA/FSF composites processed with ELO or MLO. One can observe that the neat PLA presented a relatively high mechanical strength with values of elastic modulus (E) and strength at yield ($\sigma_y$) of nearly 1200 and 65 MPa, respectively, whereas the elongation at break ($\varepsilon_b$) was 8.6%. The incorporation of the FSF fillers led to a more rigid material, increasing slightly the E value to approximately 1270 MPa and hardness from 75.8 to 76.4. However, the values of $\sigma_y$ and $\varepsilon_b$ were drastically reduced to approximately 14.7 MPa and 2.3%, respectively, whereas the impact strength was also reduced from 32.2 kJ/m$^2$ to 10.9 kJ/m$^2$. Therefore, the presence of FSF induced a considerable reduction in the mechanical ductility and toughness of PLA. This observation can be related to the low interfacial adhesion attained between the biopolyester and the lignocellulosic fillers [39,40]. This implies that the main fracture mechanism was based on filler extraction rather than filler and/or matrix fracture, which would require to dissipate more energy [41]. A similar mechanical behavior in green composites due a poor matrix-to-particle interaction was reported elsewhere [42,43]. It is also worthy to mention that injection-molded polymer composites based on fibers or long fillers typically consist of a processing-induced three-layer microstructure [44]. In the skin layers, fillers are mainly oriented parallel to the mold filling direction. As opposite, the main filler orientation in the core layer is perpendicular to the mold filling direction. Since the here-prepared pieces were relatively thick, that is, 4 mm, one can consider that most of the fillers remained mainly oriented perpendicular to the mold filling direction. This fact further supports the lower mechanical resistance properties attained in the green composite pieces when subjected to axial forces during the tensile test. However, filler orientation is expected to play a minor a role due to, as shown previously in Figure 1, the particles were predominantly round.

**Table 2.** Mechanical properties of the polylactide (PLA)/flaxseed flour (FSF) composite pieces processed with epoxidized linseed oil (ELO) and maleinized linseed oil (MLO) in terms of elastic modulus (E), strength at yield ($\sigma_y$), elongation at break ($\varepsilon_b$), impact strength, and Shore D hardness.

| Piece | Tensile Properties | | | Impact Strength (kJ/m$^2$) | Hardness |
|---|---|---|---|---|---|
| | E (MPa) | $\sigma_y$ (MPa) | $\varepsilon_b$ (%) | | |
| PLA | 1197.1 ± 20.5 | 64.9 ± 0.5 | 8.6 ± 0.2 | 32.2 ± 0.3 | 75.8 ± 1.2 |
| PLA/FSF | 1269.4 ± 26.2 | 14.7 ± 1.0 | 2.3 ± 0.3 | 10.9 ± 0.8 | 76.4 ± 2.8 |
| PLA/FSF + 2.5ELO | 1125.7 ± 20.3 | 17.9 ± 0.4 | 2.9 ± 0.3 | 11.7 ± 0.2 | 77.6 ± 2.1 |
| PLA/FSF + 5ELO | 1106.7 ± 22.7 | 18.1 ± 0.3 | 2.4 ± 0.2 | 12.3 ± 0.4 | 81.2 ± 0.4 |
| PLA/FSF + 2.5MLO | 969.6 ± 15.2 | 14.0 ± 0.7 | 7.8 ± 0.3 | 14.9 ± 0.5 | 76.6 ± 1.4 |
| PLA/FSF + 5MLO | 802.0 ± 18.9 | 12.1 ± 0.4 | 16.7 ± 0.5 | 17.1 ± 0.4 | 75.4 ± 1.9 |

In relation to the green composite pieces containing ELO, these pieces presented lower values of E but higher values of $\sigma_y$, in the range of 1100–1200 MPa and around 18 MPa, respectively, while the $\varepsilon_b$ values were also slightly higher. The impact strength and hardness values also increased, reaching values of 12–11 kJ/m$^2$ and 82–77, respectively. The two ELO-containing green composite pieces presented similar mechanical values though, in overall, the ductility improvement was low and slightly better for the composite pieces processed with 2.5 phr. Although these pieces showed a mechanical improvement with respect to the PLA/FSF composite, their properties were still inferior compared with those of the neat PLA piece, especially in terms of ductility and toughness. In this context, some authors have previously indicated that ELO can exert a plasticizing effect on polyester matrices [30,45], but this effect was not sufficient to counteract the brittleness produced by the lignocellulosic particles addition.

Regarding MLO, the addition of this multi-functionalized vegetable oil produced a similar effect than that observed for ELO but, interestingly, more intense. Then, the MLO-containing composite pieces reached E values of approximately 970 MPa and 802 MPa and $\sigma_y$ values of 14 MPa and 12.1 MPa, for contents of 2.5 phr and 5 phr, respectively. Additionally, $\varepsilon_b$ notably increased up to 7.8%, for the

pieces processed with 2.5 phr of MLO, and 16.7%, for 5 phr of MLO. In other words, with a 5 phr of MLO content, the $\varepsilon_b$ values presented a 2- and 6-fold increase in relation to the neat PLA and PLA/FSF composite, respectively. This effect can be ascribed mainly to the plasticization occurred in the PLA matrix caused by MLO [32]. Moreover, the addition of only 2.5 phr of MLO increased impact strength to 14.9 kJ/m$^2$ whereas 5 phr resulted in 17.1 kJ/m$^2$. Although these values were still far below those obtained for the neat PLA piece, they represent an increase in toughness of nearly 37% and 57%, respectively, in relation to the PLA/FSF composite piece. This result can be ascribed to the plasticizing effect of MLO on PLA by which it reduces the intermolecular forces, weakening hydrogen bonds, and then increases the free volume. Consequently, more flexible and tougher material is obtained [46]. In this regard, MLO can provide the green composite with some interface modifying properties, which has been observed previously in other green composites studies using reactive additives based on MAH groups. For instance, the impact strength of PLA/almond shell flour (ASF) composites was 65% higher after the addition of 1 phr of MLO [11]. Similar improvements in the impact strength of polyethylene/kaolin composites were reported by Hindryckx et al. [47] using petrochemical maleic anhydride-grafted polyethylene (MAH-*g*-PE). It is also worthy to note that hardness nearly remained constant, in the 77–75 range, indicating that, in addition to plasticization, the interfacial adhesion of the green composites was improved simultaneously. Similar approaches have also been obtained by pre-treatment of the filler surface with reactive coupling agents, for instance, those from the silane family [48]. The mechanical properties can be improved due to an enhanced adhesion at the biopolymer–filler interface that is produced from the specific reactive organo-functionalities of each silane that can react with some chemical groups of the biopolymer chains to form newly covalent bonds or, at least, to favor certain interaction between them.

Based on the above, one can conclude that the toughening effect of ELO on the PLA/FSF composites was significantly lower than that of MLO. Although both multi-functionalized linseed derived oils delivered a plasticizing effect on the PLA matrix, the ductility and toughness improvement was notably higher in the case of MLO. In this regard, one can consider that the different grade of solubility of ELO and MLO in the PLA matrix played an important role in the plasticization process. On the one hand, saturation of epoxidized plant-derived oils has been observed in PLA at relatively low contents, that is, ~2.5 phr [15]. On the other, MLO contents of at least 7.5 phr were needed to saturate PLA [33]. Furthermore, the multiple reactive groups present in the oils can chemically interact with the hydroxyl (–OH) and carboxyl (–COOH) terminal groups of the PLA chains, generating a macromolecule of higher $M_W$ based on a linear chain-extended, branched or even cross-linked structure, and also with the –OH groups present on the cellulose surface to form new carboxylic ester (–COO–) bonds [11,15,33]. In the case of MLO, this process is ascribed to the formation of new –COO– bonds through the reaction of the multiple maleic anhydride functionalities present in MLO with the –OH groups of both the terminal chains of PLA and cellulose on the filler surface. For ELO, ester bonds are proposed to be formed by the simultaneous reaction of the –COOH terminal groups of the PLA chains and the –OH surface groups of cellulose with some of the multiple epoxy groups of ELO. In polyesters, this is produced by a mechanism of epoxy ring-opening that results in chain extension by glycidyl esterification of the –COOH groups, which precedes hydroxyl end-group etherification. In cellulose, this reaction generates C—O—C bonds with hydroxyl side-group formation. However, the expected reactivity with polyesters of the epoxy groups present in ELO is higher than that of the MAH groups of MLO. Therefore, the resultant macromolecular structure of the green composite processed with ELO was more affected, which would explain the higher mechanical strength and lower toughness observed when epoxidized linseed oil is added instead of the maleinized one.

### 3.4. Morphology of PLA/FSF Composite Pieces

Figure 4 shows the fracture surfaces of the neat PLA piece and the green composite pieces of PLA/FSF after Charpy's impact tests taken by FESEM at 1000× magnification. In Figure 4a, corresponding to neat PLA, one can observe that the piece showed a smooth surface. This observation

indicates that the material presents a brittle fracture behavior, though certain plastic deformation can also be observed. Figure 4b shows that the incorporation of the FSF particles into the PLA matrix resulted in a remarkable change in the fracture surface of the pieces. It can be observed that a considerable amount of both small and large cavities and holes were heterogeneously distributed along the PLA matrix, presumably due to the debonding of the FSF particles from the composite piece during fracture. A large gap between the lignocellulosic fillers and the biopolyester was also noticed, confirming the lack of interaction between both composite components. This poor interfacial adhesion can cause the above-described tenacity loss during the mechanical analysis in the green composite pieces that were processed without multi-functionalized oils. One can also notice the presence of an oily coating around the FSF particles, which can be ascribed to the flaxseed oil ejected from the fillers during the fracture process. Indeed, flaxseed is known to enclose high contents of lipids, nearly 45–40%, in which around 48% correspond to unsaturated compounds such as omega-3 and omega-6 [49,50]. This ejected oil could also produce the large number of voids or bubbles observed in the fracture surface of the green composite sample due to its lower boiling and degradation temperature.

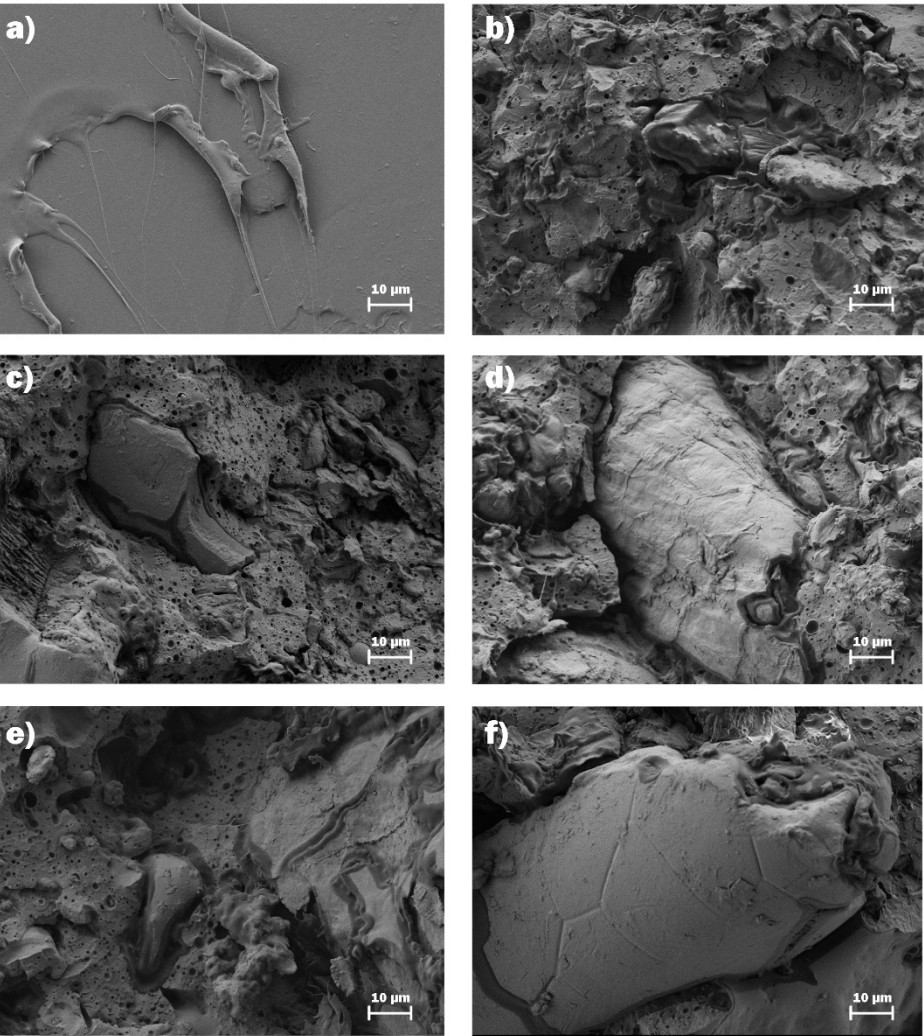

**Figure 4.** Field emission scanning electron microscopy (FESEM) images of the fracture surfaces of the polylactide (PLA)/flaxseed flour (FSF) composite pieces processed with epoxidized linseed oil (ELO) and maleinized linseed oil (MLO) of: (**a**) PLA; (**b**) PLA/FSF; (**c**) PLA/FSF + 2.5ELO; (**d**) PLA/FSF + 5ELO; (**e**) PLA/FSF + 2.5MLO; (**f**) PLA/FSF + 5MLO. Images were taken at 1000× and scale markers are of 10 μm.

Figure 4c,d shows the fracture surfaces of the PLA/FSF composite pieces processed with 2.5 phr and 5 phr of ELO, respectively. Certain interfacial improvement can be observed in the composite piece containing 2.5 phr of ELO since both the number of holes and the filler-to-matrix gaps were reduced. Although the composite piece with 5 phr of ELO showed a similar morphology, one can also observe that the previously described phenomenon of saturation of the vegetable oil occurred. This excess of plasticizer can promote the chemical interactions of the multi-functionalized vegetable oils with the biopolymer matrix. This process results in a free volume reduction that leads to an increase in mechanical resistance by a reaction of chain extension, branching and/or cross-linking, but also a matrix flexibility loss [51,52]. Other authors have reported that a plasticizer excess may cause the apparition of a separated phase rich in plasticizer, which can have a negative effect on the material ductile properties [53,54]. In any case, saturation in the PLA matrix can support the lower mechanical performance observed for the ELO-containing green composite pieces. In the case of the green composite pieces processed with MLO, shown in Figure 4e,f, one can observe that the number of cavities and holes was reduced and also the multi-functionalized oil effectively reduced and occupied the gap between the fillers and matrix. This new phase can result from the combination of the added MLO with the ejected linseed oil, which potentially favored the interfacial adhesion of the composite and, thus, supports the increase in ductility and impact strength. Previous research studies about green composites based on PLA and lignocellulosic fillers processed with MLO have shown similar morphologies, which correlates well with their enhanced ductility performance [11,12,33].

### 3.5. Thermal Properties of PLA/FSF Composite Pieces

Figure 5 gathers the DSC thermograms obtained during the second heating scan of the PLA and the PLS/FSF composites. The main thermal parameters obtained from the DSC curves are summarized in Table 3. One can observe a step change in the base line in the temperature range between 55–65 °C, which corresponds to the PLA's glass transition temperature ($T_g$). For the neat PLA, the mean $T_g$ value was located at 63.2 °C. The exothermic peak observed between 100 °C and 130 °C corresponds to the cold crystallization temperature (Tcc), which was centered at 119.4 °C for the unfilled PLA sample. Finally, the endothermic peaks comprised between 160 °C and 180 °C correspond to the melting process of the total crystalline fraction in PLA. The neat PLA sample showed two overlapped peaks during melting, that is, the first melting temperature ($T_m$) occurred at 167.6 °C whereas the second one was observed at 172.2 °C. This double-melting peak phenomenon can be ascribed to the formation of crystalline structures in polyesters with dissimilar lamellae thicknesses or the presence of crystallite blocks with different degrees of perfection [55]. A similar effect was described for PLA by Yasuniwa et al. [56] who ascribed this phenomenon to either the existence of polymorphic forms or melting during heating of a metastable crystalline form that recrystallizes immediately into a more stable one. One can observe that the incorporation of the FSF particles produced a reduction of both the $T_g$ and $T_{CC}$ values to 61.9 °C and 108.8 °C, respectively. The slight reduction observed for the glass transition region can be related to the released oil from the fillers whereas the shift to lower temperatures for cold crystallization suggests that the FSF particles can contribute to a heterogeneous crystallization process by acting as external nuclei for PLA crystallization. Indeed, lignocellulosic microparticles are known to act as a nucleating agent that contributes to increasing the crystallinity levels of PLA [57,58]. The crystallinity degree also increased from 9.7% to 10.9%. Other research studies devoted to green composites have reported similar findings. For instance, the addition of 2.5 wt% ASF increased PLA's $X_C$ from 9.2% to 11.9% [11], whereas 20% HSF increased it from 9.6% to 14.8% [30]. It is also worthy to mention that the lignocellulosic fillers suppressed the double-melting peak phenomenon of PLA and the biopolymer melted in a single peak at 170.5 °C. This observation further confirms that the lignocellulosic particles favored the formation of more homogenous crystalline structures or crystallite blocks with similar degrees of perfection [59,60].

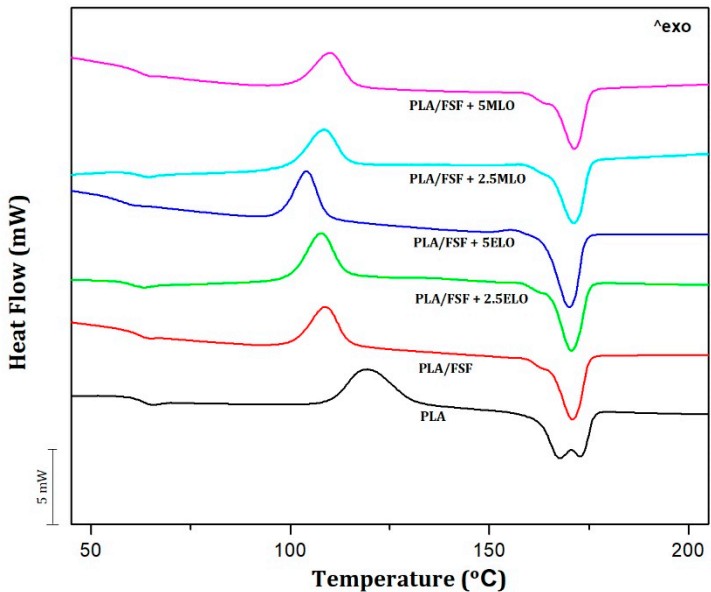

**Figure 5.** Differential scanning calorimetry (DSC) thermograms during second heating corresponding to the polylactide (PLA)/flaxseed flour (FSF) composite pieces processed with epoxidized linseed oil (ELO) and maleinized linseed oil (MLO).

**Table 3.** Thermal properties of the polylactide (PLA)/flaxseed flour (FSF) composite pieces processed with epoxidized linseed oil (ELO) and maleinized linseed oil (MLO) in terms of glass transition temperature ($T_g$), cold crystallization temperature ($T_{CC}$), melting temperature ($T_m$), cold crystallization enthalpy ($\Delta H_{CC}$), melting enthalpy ($\Delta H_m$), and degree of crystallinity ($X_c$).

| Piece | $T_g$ (°C) | $T_{cc}$ (°C) | $T_m$ (°C) | $\Delta H_{CC}$ (J/g) | $\Delta H_m$ (J/g) | $X_c$ (%) |
|---|---|---|---|---|---|---|
| PLA | 63.2 ± 0.6 | 119.4 ± 0.9 | 167.6 ± 1.1/172.2 ± 1.2 | 27.3 ± 0.4 | 36.4 ± 0.5 | 9.7 ± 0.2 |
| PLA/FSF | 61.9 ± 0.8 | 108.8 ± 1.3 | 170.5 ± 1.5 | 18.7 ± 0.6 | 25.9 ± 0.8 | 10.9 ± 0.3 |
| PLA/FSF + 2.5ELO | 60.5 ± 0.7 | 107.8 ± 0.7 | 170.4 ± 1.5 | 21.5 ± 0.2 | 26.1 ± 0.1 | 7.1 ± 0.4 |
| PLA/FSF + 5ELO | 57.4 ± 1.1 | 103.9 ± 0.9 | 169.5 ± 1.7 | 17.2 ± 0.5 | 26.3 ± 0.3 | 13.8 ± 0.2 |
| PLA/FSF + 2.5MLO | 62.6 ± 0.9 | 108.6 ± 1.1 | 171.0 ± 1.6 | 18.9 ± 0.5 | 24.7 ± 0.6 | 8.8 ± 0.2 |
| PLA/FSF + 5MLO | 62.2 ± 0.8 | 109.9 ± 1.4 | 174.9 ± 1.7 | 18.7 ± 0.6 | 25.7 ± 0.4 | 10.7 ± 0.4 |

In relation to the multi-functionalized linseed derived oils, it can be observed that ELO produced a reduction of the $T_g$ and $T_{CC}$ values and increased considerably the percentage of crystallinity of PLA whereas MLO, in general, had a negligible effect on the thermal properties of the PLA/FSF composites. In particular, $T_g$ was reduced to 60.5 °C after the addition of 2.5 phr ELO, and 57.4 °C for 5 phr ELO. This is in agreement with the above-described plasticizing effect that ELO produces on the PLA matrix by which it increases free volume, favor chains mobility, and also enables the crystallization process to occur earlier [61,62]. A similar reduction in the $T_g$ values has been achieved in other PLA-based composites using ELO. For instance, Balart et al. [30] showed a reduction in the $T_g$ value of approximately 8 °C for PLA/HSF composites after the addition of 22.5 phr ELO. Similarly, Alam et al. [63] reported a $T_g$ reduction of around 10 °C when 15% ELO was added to PLA/multi-walled carbon nanotubes (MWCNTs) nanocomposites. The composite sample containing 5 phr of ELO also underwent cold crystallization at a significant lower temperature, that is, the $T_{CC}$ value was 103.9 °C, and it also increased the total crystalline fraction of PLA to 13.8%. Therefore, ELO played the role of a plasticizer in PLA, promoting chain mobility and allowing the arrangement of their movement into more packed forms [32]. As opposite, the $T_g$ and $T_{CC}$ values of the MLO-containing composite samples were relatively similar to those of the untreated PLA/FSF composite. This result confirms that, for the here-tested contents, MLO was mainly located in the matrix-to-filler interface as a reactive compatibilizer and then exerted a minor plasticizing effect, which was previously observed during both the mechanical and FESEM analyses. No further relevant changes on the melting profile of the

treated composites were observed and all the samples melted in a single peak located at 171 °C, for the green composite piece processed with 2.5 phr of MLO, and ~175 °C, for 5 phr of MLO, with both showing $X_C$ values in the 11–10% range.

Thermal degradation of the FSF particles, neat PLA, and the PLA/FSF composites processed without and with ELO and MLO was studied by TGA. The effect of temperature on mass is plotted in Figure 6 and the results are summarized in Table 4. Figure 6a gathers the TGA curves, while the DTG curves are included in Figure 6b. Three main weight losses can be observed for FSF. Initially, a mass loss of approximately 3% can be seen around 100 °C. This peak mainly related to the water release from the lignocellulosic fillers, which are very hydrophilic materials. Then, one can observe that a sharp mass loss from 230 °C to 370 °C took place. The mass loss associated with this peak was ~52% and it corresponds to the "active pyrolysis zone". In this region, the mass loss rate is high and it comprises the major devolatilization step of biomass pyrolysis. In particular, this step has been mainly associated with the thermal decomposition of hemicellulose and other low-$M_W$ components in lignocellulose, followed by cellulose degradation [64]. The third and last weight loss was regarded as a gradual degradation with a slower mass loss rate, which is ascribed to the "passive pyrolysis zone". This thermal loss resulted in a mass loss of ~33% and it started at approximately 410 °C and ended at nearly 650 °C. This mass loss relates to char decomposition and degradation of the remaining lignin [64]. One can also observe that, in both the TGA and DTG curves, the end of the second degradation peak overlapped with the initiation of the third one so that the latter peak was observed as a tailing. At higher temperatures, that is, above 650 °C, the thermal degradation continued progressively. However, since most volatiles were already pyrolyzed, the mass loss rate was no longer significant. The residual mass was nearly 12% and it can be related to the mineral content in flaxseed, which has been previously reported to range at 15–10% [49].

The TGA curve of neat PLA shows two degradation stages for the biopolyester. From about 335 °C to 390 °C, the first and main degradation step took place, showing a mass loss of ~95%. This peak was sharp due to it relates to chain-scission reaction of the PLA macromolecules into smaller chain fragments through the breakage of its ester groups [65]. Then, a second low-intensity mass loss (<4%) was attained in the DTG curve of PLA from approximately 405 °C to 450 °C. The latter step is ascribed to the thermal degradation of the remaining small-$M_W$ PLA fragments produced during the previous degradation process. The incorporation of FSF reduced the onset degradation temperature, measured at the mass loss of 5% ($T_{5\%}$), by approximately 60 °C and 44 °C, respectively, and also the degradation temperature ($T_{deg}$). It is also worth to mention that the fillers generated an additional degradation peak in the thermal range from 365 °C to 590 °C, which can be related to the thermal decomposition of FSF. The resultant thermal stability impairment has been recently ascribed to the low thermal stability of the lignocellulosic fillers and also to the remaining moisture, which is extremely difficult to remove [12,66]. Moreover, the thermal stability of PLA can also be influenced negatively by the high shear and frictional forces experienced during melt extrusion [12]. Similar results were obtained for other green composites based on flaxseed by Samal et al. [67].

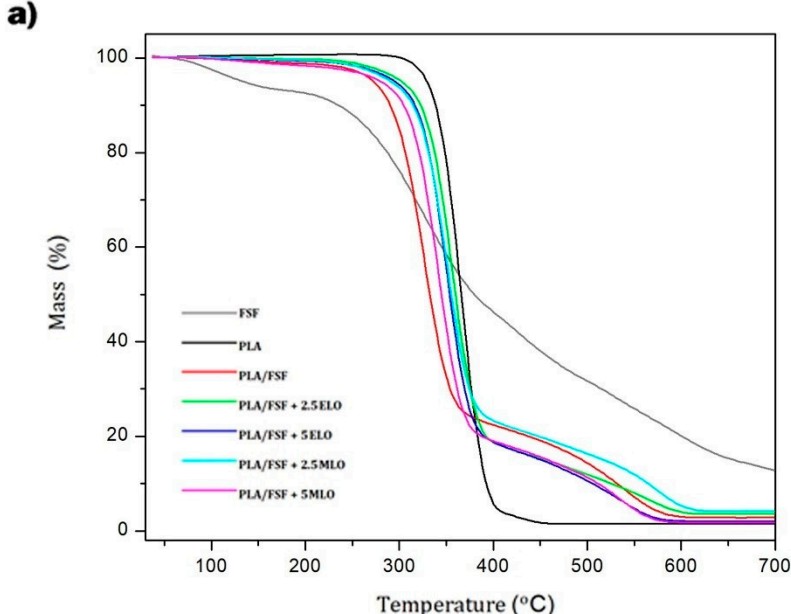

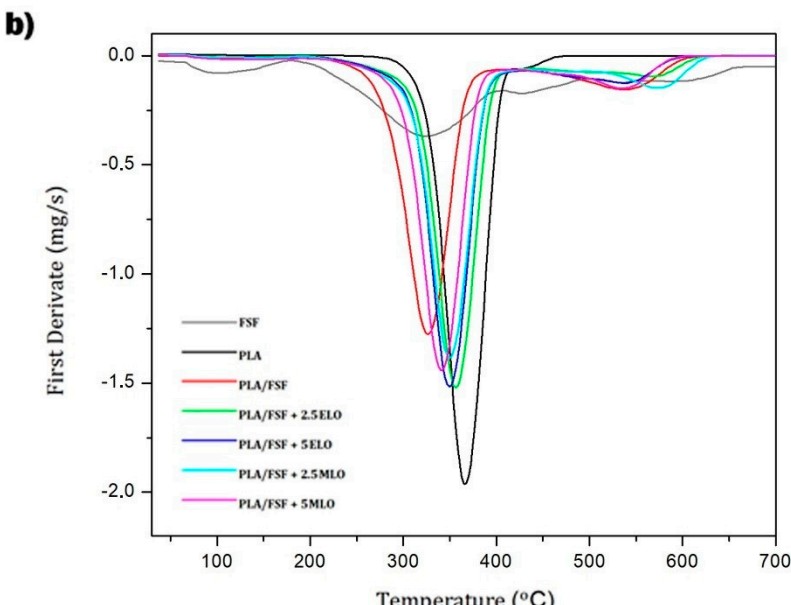

**Figure 6.** (**a**) Thermogravimetric analysis (TGA) and (**b**) first derivate thermogravimetric (DTG) curves corresponding to the polylactide (PLA)/flaxseed flour (FSF) composite pieces processed with epoxidized linseed oil (ELO) and maleinized linseed oil (MLO).

**Table 4.** Main thermal parameters of the polylactide (PLA)/flaxseed flour (FSF) composite pieces processed with epoxidized linseed oil (ELO) and maleinized linseed oil (MLO) in terms of onset temperature of degradation ($T_{5\%}$), degradation temperature ($T_{deg}$), and residual mass at 700 °C.

| Piece | $T_{5\%}$ (°C) | $T_{deg1}$ (°C) | $T_{deg2}$ (°C) | $T_{deg3}$ (°C) | Residual Mass (%) |
|---|---|---|---|---|---|
| FSF | 134.8 ± 0.4 | 102.1 ± 0.3 | 221.1 ± 0.2 | 409.8 ± 0.3 | 12.81 ± 0.4 |
| PLA | 336.9 ± 0.5 | - | 373.4 ± 1.5 | 405.1 ± 1.2 | 1.51 ± 0.2 |
| PLA/FSF | 277.0 ± 0.3 | - | 329.4 ± 1.1 | 367.9 ± 1.2 | 2.82 ± 0.2 |
| PLA/FSF + 2.5ELO | 304.9 ± 0.5 | - | 359.9 ± 0.8 | 391.2 ± 0.9 | 3.62 ± 0.3 |
| PLA/FSF + 5ELO | 295.6 ± 0.2 | - | 352.8 ± 1.4 | 379.5 ± 0.8 | 2.96 ± 0.2 |
| PLA/FSF + 2.5MLO | 291.0 ± 0.4 | - | 353.4 ± 0.7 | 384.2 ± 1.1 | 4.22 ± 0.3 |
| PLA/FSF + 5MLO | 285.1 ± 0.4 | - | 347.5 ± 0.8 | 374.8 ± 0.9 | 3.81 ± 0.3 |

Interestingly, the addition of both multi-functionalized oils derived from linseed provided an enhancement of the thermal stability of the PLA/FSF composites. In particular, the $T_{5\%}$ values shifted from approximately 277 °C, for the green composite processed without multi-functionalized vegetable oils, to values in the 305–295 °C range. The highest improvement was observed for the ELO-containing PLA/FSF composites. Similarly, the $T_{deg}$ values increased from 329 °C for the green composite processed without oils to values in the 360–347 °C range, which are certainly close to that of neat PLA. The positive effect of ELO on the thermal stability of PLA-based materials was previously studied by Balart et al. [30] who showed an increase of the thermal stability at high ELO contents. In the case of MLO, our recent research works reported that this multi-functionalized vegetable oil can also exert a positive influence on the overall thermal stability of green composites based on PLA [12,33]. This effect was particularly related to the improved interfacial adhesion of the composite. Therefore, the thermal stability enhancement is directly related to the chemical interaction achieved by the reactive oils due to the covalent bonds established between the lignocellulosic fillers and the biopolymer matrix. The higher thermal stability observed for ELO can be ascribed to both its higher reactivity and lower solubility in the PLA matrix. The latter phenomenon is based on the fact that the non-reacted epoxy groups could successfully provide a thermal oxidation effect, while the remaining oil could also produce a physical barrier that obstructs the removal of volatile products produced during thermal decomposition. It can also be observed that all the green composites reached amounts of residual mass in the 4–3% range at 700 °C, which can be ascribed to the formed char from PLA and, more noticeably, from FSF.

*3.6. Thermomechanical Properties of PLA/FSF Composite Pieces*

In Figure 7 the evolution of the storage modulus and the damping factor (*tan δ*) as a function of increasing temperature are comparatively plotted for the neat PLA piece and its green composite pieces with FSF processed ELO and MLO. The values of the storage modulus measured at 40 °C and 110 °C and $T_g$ are shown in Table 5. The storage modulus relates to the stiffness of the pieces since it is based on the stored elastic energy of the material. It can be seen in Figure 7a that the storage modulus values of PLA, as a typical thermoplastic, were relatively high at low temperatures and then sharply decreased when the glass transition was exceeded at nearly 65 °C. This thermomechanical change is related to the mobility increment of the PLA amorphous phase. In addition, one can also observe that the storage modulus values increased from approximately 85 °C to 95 °C due to the above-reported cold crystallization phenomenon. Similar thermomechanical curves were obtained for the PLA/FSF composite pieces, however, the lignocellulosic particles induced an overall increase in stiffness. For instance, the storage modulus at 40 °C increased from nearly 567 MPa, for the neat PLA piece, to approximately 1509 MPa, for the PLA/FSF piece. The fillers' presence additionally reduced the initiation of the glass transition region and the cold crystallization process also shifted to lower temperatures.

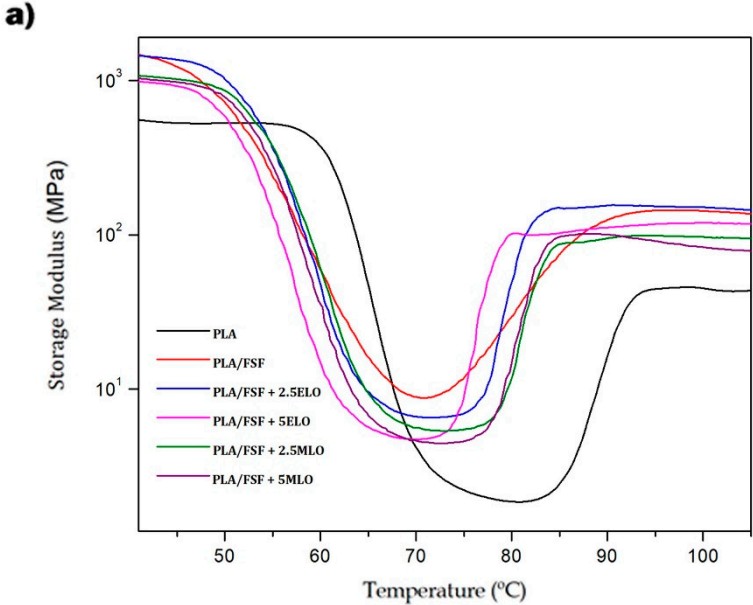

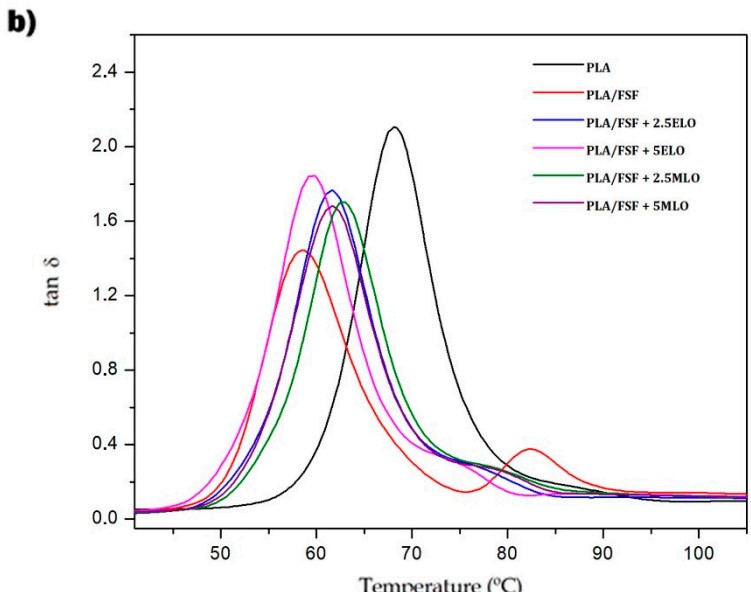

**Figure 7.** Evolution of the (**a**) storage modulus and (**b**) dynamic damping factor (*tan δ*) as a function of temperature for the polylactide (PLA)/flaxseed flour (FSF) composite pieces processed with epoxidized linseed oil (ELO) and maleinized linseed oil (MLO).

Regarding ELO and MLO, the composite pieces processed with both multi-functionalized oils derived from linseed showed similar thermomechanical curves than the composite piece processed without oil but with lower storage thermomechanical values at the highest oil contents. This effect, which can be ascribed to the plasticizing role of both vegetable oils in the PLA matrix [30], then led to a softer material. For instance, at 110 °C, the values of the storage modulus decreased from 130.4 MPa for the untreated composite piece to 112.8 MPa and 75.2 MPa for the 5-phr ELO- and MLO-containing composite pieces, respectively. It is also worthy to note that at low temperatures, for the lowest vegetable oil content, the storage values were similar or even slightly higher than those of the green composite processed without oil. This observation suggests that the PLA matrix was not saturated at 2.5 phr. Therefore, the interfacial adhesion enhancement predominated over plasticization and then

both multi-functionalized vegetable oils promoted an enhancement in the mechanical strength of the green composites.

**Table 5.** Main thermomechanical parameters of the polylactide (PLA)/flaxseed flour (FSF) composite pieces processed with epoxidized linseed oil (ELO) and maleinized linseed oil (MLO) in terms of storage modulus measured at 40 °C and 110 °C and glass transition temperature ($T_g$).

| Piece | Storage Modulus (MPa) | | $T_g$ (°C) |
|---|---|---|---|
| | 40 °C | 110 °C | |
| PLA | 566.7 ± 9.4 | 47.6 ± 3.9 | 68.4 ± 3.6 |
| PLA/FSF | 1509.5 ± 10.1 | 130.4 ± 4.2 | 58.5 ± 2.8 |
| PLA/FSF + 2.5ELO | 1466 ± 9.2 | 137.1 ± 4.6 | 61.3 ± 4.1 |
| PLA/FSF + 5ELO | 999.3 ± 9.8 | 112.8 ± 3.8 | 59.4 ± 3.7 |
| PLA/FSF + 2.5MLO | 1087.5 ± 10.3 | 93.5 ± 4.5 | 63.2 ± 3.5 |
| PLA/FSF + 5MLO | 1046.2 ± 9.9 | 75.2 ± 3.9 | 61.7 ± 2.3 |

Figure 7b shows the evolution of *tan δ* versus temperature for the PLA and PLA/FSF composites processed without and with ELO and MLO. This thermomechanical property relates, in a cyclic deformation, the ratio of the energy lost to the energy stored due to the viscous and elastic behaviors. The α-relaxation of PLA was seen as the intense peak located between 55 °C and 85 °C. This thermomechanical property is related to the biopolymer's $T_g$. One can observe that in the case of the neat PLA piece this peak was centered at 68.4 °C, which is slightly higher than that observed during DSC analysis. This difference can be explained by variances in the sample crystallinity and test conditions [12]. It can also be observed that the FSF introduction shifted the PLA's α-relaxation peak to 58.5 °C and it additionally reduced the peak intensity. This observation indicates that the relaxation of the PLA chains was partially suppressed by the fillers' presence [68]. Therefore, the green composite piece developed lower crystallinity since the number of molecules that underwent glass transition was reduced, which is consistent with the previous DSC analysis. Slightly higher $T_g$ values and also more intense peaks were obtained for the ELO- and MLO-containing composite pieces, particularly at the lowest oil contents, which indicate that the achieved interfacial adhesion reduced the thermomechanical effect induced by the FSF particles. Similar results were obtained for PLA/silica ($SiO_2$) composites processed with the linseed derived chemically modified oils, showing that the storage modulus of the composite with 5 phr of MLO was reduced from 830 MPa to 600 MPa and this reduction was maintained with the oil content increase [69].

*3.7. Water Uptake of PLA/FSF Composite Pieces*

The evolution of the water uptake of the neat PLA piece and the PLA/FSF composite pieces processed without and with linseed derived oils is plotted in Figure 8. Water uptake was evaluated during an immersion time of 85 days at room temperature. Table 6 gathers the values of $W_s$, D, and $D_c$. It can be observed that the weight gains of all the here-studied PLA pieces due to water sorption versus the immersion time followed the Fick's first law. Briefly, all the materials presented an initial stage based on a rapid increment in water uptake followed by a second stage in which mass tended asymptotically toward saturation, that is, they reached $W_s$, after approximately 15 days of water immersion. The lowest water uptake values were observed for the unfilled PLA piece with a $W_s$ value of ~0.8 wt%. The incorporation of the FSF particles produced a remarkable increase in the amount of adsorbed water, reaching a $W_s$ value of ~5.4 wt%. This is a recurrent technical issue faced in composite materials based on lignocellulosic fillers since the cellulose and hemicellulose present in the particles contain a large number of –OH groups that highly interact with water molecules [70,71]. The equilibrium water uptake increased significantly in the green composite pieces processed with ELO or MLO and it was found to depend on the type of oil but, interestingly, not on the oil content. In particular, the green composite pieces containing ELO entrapped more water, reaching $W_s$ values of

nearly 20 wt% and 18.1 wt% for contents of 2.5 phr and 5 phr, respectively. The MLO-containing green composites pieces processed with 2.5 phr and 5 phr showed similar adsorption curves but with lower $W_s$ values, that is, approximately 16.7 wt% and 17.1 wt%, respectively. This result is related to the free volume enlargement attained in the PLA matrix, favoring water diffusion, due to the plasticizing effect of the multi-functionalized vegetable oils. Furthermore, the formation of cavities and holes in the PLA matrix, previously observed in the fracture surface morphology, could further favor the water uptake phenomenon in these green composite pieces. The lower water uptake attained for the pieces containing 5 phr of ELO can be ascribed to the aforementioned interfacial adhesion increase, which led to a composite structure based on a PLA matrix containing an ELO-rich separated phase. In relation to MLO, lower $W_s$ values were observed due to the fact that the vegetable oil was placed at the filler-to-matrix interface and also a lower number of cavities were formed. Similar results were previously observed for MLO-containing green composites with 40% of walnut shell flour (WSF) in which a decrease in water uptake from approximately 35 wt% to 20 wt% was attained after the addition of MLO [12].

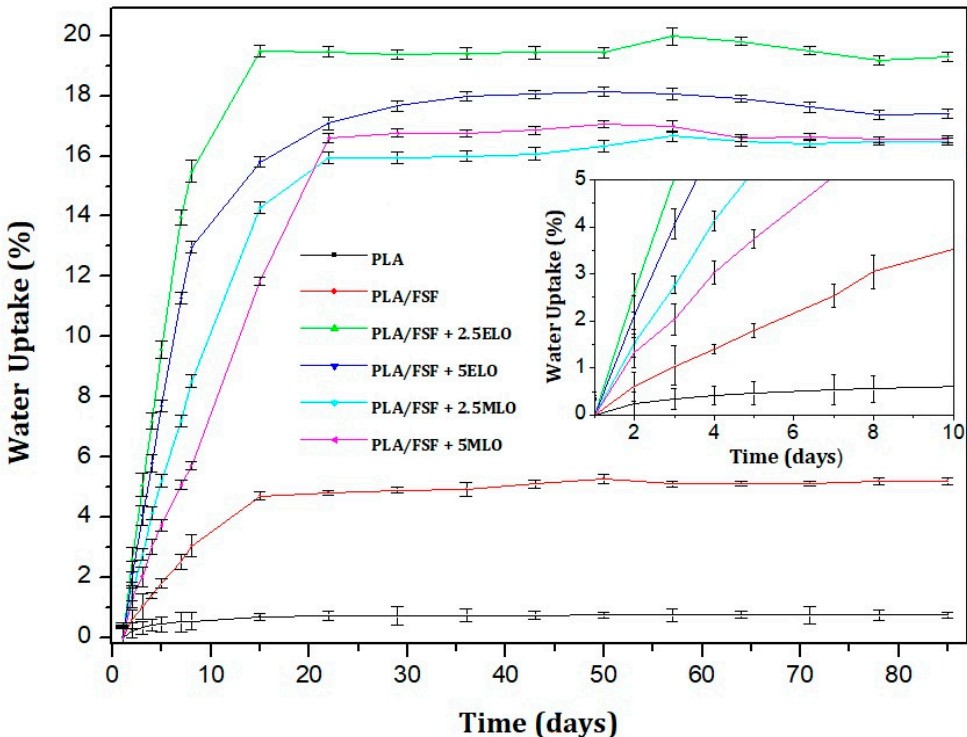

**Figure 8.** Water uptake of the polylactide (PLA)/flaxseed flour (FSF) composite pieces processed with epoxidized linseed oil (ELO) and maleinized linseed oil (MLO).

Finally, the diffusion properties of the pieces immersed in distilled water were also estimated according to Fick's law, as described above in Equation (5), by measuring the slope of the first part of the weight gain curve versus square root of time. Figure 9 shows the $W_t/W_s$ ratio as a function of the square root of time for the injection-molded pieces of PLA and PLA/FSF processed without and with ELO and MLO. A Fickian-like or pseudo-Fickian behavior was observed for the PLA pieces over an initial period. However, at longer times, the non-equilibrium state of the glassy region relaxed and allowed more water to enter into the biopolymer gradually over the dwelling time [72]. A lower slope in the $W_t/W_s$ ratio versus square root of time graph means a lower diffusion coefficient, which represents a lower speed in terms of water sorption. One can observe that the Dc value of the neat PLA piece was 34.62 × $10^{-9}$ cm$^2$/s, which was very similar to the coefficient reported earlier by Siparsky et al. [73] at 25 °C. The latter value was 20–30 times higher than those observed for the green composite pieces. This result

points out that the FSF presence reduces the diffusion of the water molecules through the PLA matrix by a tortuous mechanism, suggesting that the water gain in the green composite is mainly determined by the entrapment of water in the lignocellulosic fillers. In this regard, it is worthy to mention that the lowest $D_c$ values were observed for the MLO-containing composite pieces, which can be ascribed to the enhanced interfacial adhesion of the composites that prevented the diffusion of water into the FSF fillers. Therefore, although the use of the here-prepared green composites in a damp atmosphere is certainly restricted, it is expected that the fillers presence and particularly the co-addition of MLO can successfully reduce water diffusion if the pieces are exposed to water for short exposure periods.

**Table 6.** Water uptake properties of the polylactide (PLA)/flaxseed flour (FSF) composite pieces processed with epoxidized linseed oil (ELO) and maleinized linseed oil (MLO) in terms of saturation weight ($W_s$), diffusion coefficient (D), and corrected diffusion coefficient ($D_c$).

| Piece | $W_s$ (wt%) | $D \times 10^9$ (cm²/s) | $D_c \times 10^9$ (cm²/s) |
|---|---|---|---|
| PLA | $0.77 \pm 0.19$ | $34.62 \pm 0.23$ | $16.44 \pm 0.11$ |
| PLA/FSF | $5.37 \pm 0.23$ | $2.51 \pm 0.14$ | $1.23 \pm 0.15$ |
| PLA/FSF + 2.5ELO | $19.97 \pm 0.17$ | $2.75 \pm 0.29$ | $1.35 \pm 0.26$ |
| PLA/FSF + 5ELO | $18.14 \pm 0.12$ | $2.31 \pm 0.16$ | $1.09 \pm 0.16$ |
| PLA/FSF + 2.5MLO | $16.67 \pm 0.09$ | $1.79 \pm 0.08$ | $0.87 \pm 0.22$ |
| PLA/FSF + 5MLO | $17.07 \pm 0.18$ | $1.24 \pm 0.24$ | $0.61 \pm 0.19$ |

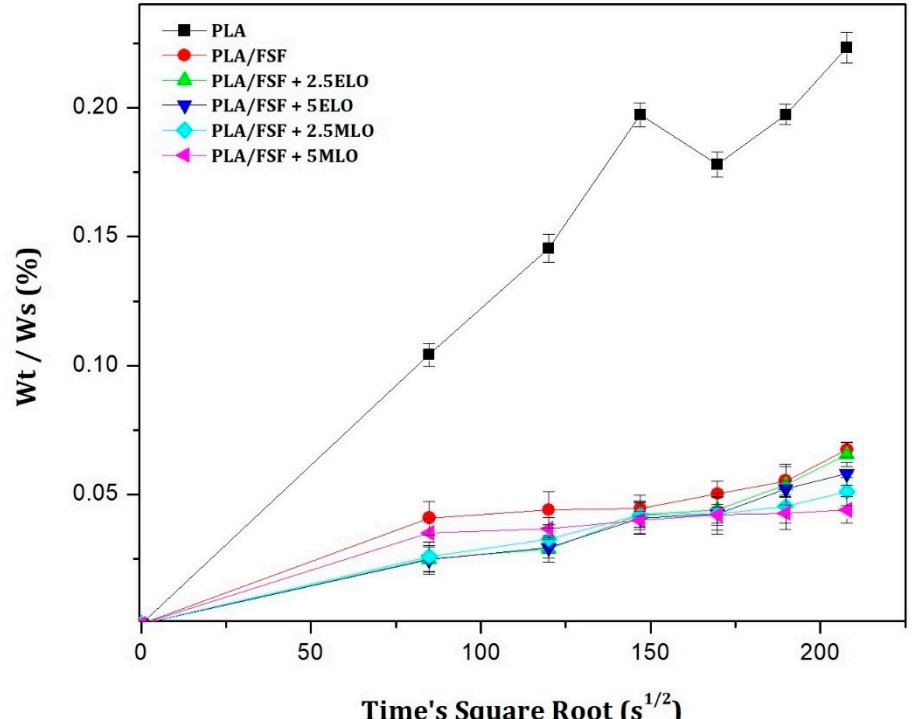

**Figure 9.** Evolution of weight sample ($W_t$)/saturation weight ($W_s$) ratio versus square root of time of the polylactide (PLA)/flaxseed flour (FSF) composite pieces processed with epoxidized linseed oil (ELO) and maleinized linseed oil (MLO).

## 4. Conclusions

Novel green composite pieces composed of PLA and FSF were developed by twin-screw extrusion and subsequent injection molding. The incorporation of the FSF fillers into PLA led to a rigidity increase and the pieces were also significantly less ductile and more brittle. The tenacity loss observed during the mechanical analysis was related to the high filler content and also to the absence or poor interfacial adhesion between both components of the green composite. This observation was supported during the

morphological analysis since the fracture surface of the green composites showed a large gap between the lignocellulosic fillers and the biopolyester matrix and also several cavities due to particle debonding during fracture. Future research works should be thus addressed to optimize the waste derived filler content by means of new methodologies such as the Kalman filter algorithm [74] or the Taguchi method [75], whereas the surface treatment of the fillers can also be explored. Interestingly, the addition during melt compounding at 2.5 phr or 5 phr of composite of the two tested multi-functionalized oils derived from linseed successfully improved the toughness and thermal stability of the pieces. It was observed that both multi-functionalized linseed derived oils delivered a plasticizing effect on the PLA matrix. However, the ductility and toughness improvement was notably higher in the case of MLO, which was ascribed to its higher solubility in the PLA matrix. Furthermore, the multiple reactive groups present in the chemically modified oils could react during extrusion with the –OH and –COOH terminal groups of the PLA chains and also with the –OH groups present on the cellulose surface. The lower mechanical and thermomechanical strength but higher toughness attained in the case of the MLO-containing green composite pieces was also supported as a result of the lower reactivity of the multiple MAH groups presents in MLO compared with the epoxy groups of ELO. Finally, although it was observed that the incorporation of the FSF particles produced a remarkable increase in the amount of entrapped water, diffusion was reduced for the green composites processed with the multi-functionalized oils derived from linseed. This improvement was related to the fact that water uptake was found to occur mainly on the lignocellulosic fillers so, even though the PLA matrix was plasticized, the oil interface successfully reduced water diffusion in the green composites.

The here-developed green composite pieces can represent a sustainable and cost-effective solution of general interest to the plastic industry according to the principles of the Circular Economy and Bioeconomy, which currently demand high-performing bio-based and biodegradable materials with a lower carbon footprint and at competitive prices. Illustrative potential uses of the materials developed herein include food trays, bottles, and caps for rigid packaging and also wood-like disposable articles for food applications such as cutlery and straws.

**Author Contributions:** Conceptualization was carried out by D.G.-S., S.T.-G., and O.F.; methodology by Á.A.; validation and formal analysis, Á.A. and S.T.-G.; investigation and data curation, Á.A. and D.L.; writing—original draft preparation, Á.A.; writing—review and editing, S.T.-G.; supervision and project administration, D.G.-S., S.T.-G., and O.F. All authors have read and agreed to the published version of the manuscript.

**Funding:** This research work was funded by the Spanish Ministry of Science, Innovation, and Universities (MICIU) project numbers RTI2018-097249-B-C21 and MAT2017-84909-C2-2-R.

**Acknowledgments:** D.L. thanks Universitat Politècnica de València (UPV) for the grant received through the PAID-01-18 program. S.T.-G. is recipient of a Juan de la Cierva—Incorporación contract (IJCI-2016-29675) from MICIU.

**Conflicts of Interest:** The authors declare no conflict of interest.

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
