# Peer review of "Valorization of Linen Processing By-Products for the Development of Injection-Molded Green Composite Pieces of Polylactide with Improved Performance"

_sustainability, doi:10.3390/su12020652_

Round 1
Reviewer 1 Report
In the proposed manuscript the authors describe the results obtained in the development of fully green composites based on a bio-based and biodegradable polyester with fillers and additives derived from flax processing by-products and waste. The results are presented in a clear manner, controls were performed, and the relevant mechanical properties and thermal behavior have been determined. In my opinion the work is relevant and timely and will be of interest to researchers in the field. However, and as the authors know, one of the challenges in the development of new bio-based composite materials is the fibre - matrix adhesion, which is of the most importance to the final properties of the composite. In this context perhaps it makes sense to add a small discussion about this topic and justify why the FSF were used without any surface treatment.
Author Response
In the proposed manuscript the authors describe the results obtained in the development of fully green composites based on a bio-based and biodegradable polyester with fillers and additives derived from flax processing by-products and waste. The results are presented in a clear manner, controls were performed, and the relevant mechanical properties and thermal behavior have been determined. In my opinion the work is relevant and timely and will be of interest to researchers in the field. However, and as the authors know, one of the challenges in the development of new bio-based composite materials is the fibre - matrix adhesion, which is of the most importance to the final properties of the composite. In this context perhaps it makes sense to add a small discussion about this topic and justify why the FSF were used without any surface treatment.
Thank you for reviewing our manuscript. As kindly suggested, we have added a brief discussion about the filler surface treatment as another potential strategy to improve the interfacial adhesion of the composite. Please see page 9, lines 318-323.
Reviewer 2 Report
This work reported a green composite that contains PLA and fillers obtained as by-product and waster form the linen industry. The mechanical properties and thermal properties of the green composite were studied through experiments. Overall, the topic is interesting and suitable for the journal. The manuscript was well prepared with some interesting results presented. It can be acceptable for publication after fully addressing the following issues.
The scientific values and novelty of the current work should be further discussed. The composite materials with PLA and bio-degradable additives have been reported in some existing work. What are the limitations of the current study? What is the influence of the manufacturing process on the mechanical and thermal properties of the green composition? For example, additive manufacturing rather than injection modeling. How did the authors determine the weight percentages for the fillers and additives? The rule for the design of the experiment should be enforced. The optimal ratio of filler to matrix should be discussed with a brief review of optimization methodologies such as Kalman filter, Taguchi method, etc. Please review the following references. https://doi.org/10.1007/s00170-018-2508-6; https://doi.org/10.1016/j.conbuildmat.2015.01.039; etc. It is interesting to include a discussion of the manufacturability of the green composition in injection modeling. For example, the quality and defects of the products, the cost of the process. What are the future works? Will real products be made for mechanical and thermal testing? Please provide high-resolution figures. For example, figure 1 and figure 7. Please consider adding error bars to the measurement of modulus. Did the authors conduct multiple experiments? The orientation and distribution of the fillers might significantly affect the mechanical properties.Author Response
This work reported a green composite that contains PLA and fillers obtained as by-product and waster form the linen industry. The mechanical properties and thermal properties of the green composite were studied through experiments. Overall, the topic is interesting and suitable for the journal. The manuscript was well prepared with some interesting results presented. It can be acceptable for publication after fully addressing the following issues.
Q1) The scientific values and novelty of the current work should be further discussed. The composite materials with PLA and bio-degradable additives have been reported in some existing work. What are the limitations of the current study?
A1) Thank you for reviewing our manuscript. We have tried to remark the main scientific achievements and novelty in the manuscript, particularly in the abstract and conclusion sections. The materials produced are mainly limited by the amount of lignocellulosic filler that can be included, which is similar to most other green composites, and obviously the resultant properties are also related to the amount of filler added. In the manuscript it is clearly indicated that it is observed a decrease in the performance when the fillers derived from the flax waste are incorporated. However, the use of the tested multi-functionalized oils, also derived from the linen industry, could successfully counteract this negative effect and produce valuable products.
Q1) What is the influence of the manufacturing process on the mechanical and thermal properties of the green composition? For example, additive manufacturing rather than injection modeling.
A2) We have added some comments about the manufacturing process of the green composites and the visual aspect of the pieces. Please see page 7, lines 237-253. About additive manufacturing, it can be feasible but we have not tested and it is difficult to ascertain the effect of the high filler content on 3D printing.
Q3) How did the authors determine the weight percentages for the fillers and additives? The rule for the design of the experiment should be enforced. The optimal ratio of filler to matrix should be discussed with a brief review of optimization methodologies such as Kalman filter, Taguchi method, etc. Please review the following references. https://doi.org/10.1007/s00170-018-2508-6; https://doi.org/10.1016/j.conbuildmat.2015.01.039; etc.
A3) The raw materials were basically weighed prior to manufacturing whereas the filler-to-matrix ratio was selected based on the best results of previous studies. Please see details in section 2.2. We have also included the suggested papers in the Conclusion section as potential future works that can be applied to optimize the filler content in these or similar green composites. Please see page 19, lines 637-640.
Q4) It is interesting to include a discussion of the manufacturability of the green composition in injection modeling. For example, the quality and defects of the products, the cost of the process. What are the future works? Will real products be made for mechanical and thermal testing?
A4) We have also added a brief discussion about the potential manufacturing of the pieces by injection molding. For this, a new section 3.2. was created. The future use of the injection-molded pieces was also briefly discussed in page 7, lines 253-255. Finally, an image showing the pieces was also included, please see new Figure 3 also in page 7.
Q5) Please provide high-resolution figures. For example, figure 1 and figure 7. Please consider adding error bars to the measurement of modulus. Did the authors conduct multiple experiments? The orientation and distribution of the fillers might significantly affect the mechanical properties.
A5) We have improved the resolution of Figures 1 and 7. Mechanical modulus values shown in Table 2 already include their standard deviation. In the case of the DMTA curves, it was not possible to include error bars since each repetition produces a new different curve. The effect of filler orientation was discussed in page 8, lines 274-283.
Reviewer 3 Report
The manuscript under the title: “Valorization of Linen Processing By-Products and Wastes for the Development of Injection-Molded Green Composite Pieces of Polylactide with Improved Performance” is appropriate for Sustainability journal. The authors present original research works, supported by experiment. The aim of article is precisely formulated. The research methodology is clearly described. The authors applied wide range of experimental research. The topic of the article is up-to-date. The organization of the article is appropriate. The abstract presents all required information and it is sufficiently informative. Overall, the paper is well prepared. Neverthless, some improvements could be applied:
line: 39-43 - please rewrite this sentence, line: 62 - both version are correct, but "the Ukraine" is most common, line: 114, 148, 152 and 191 - please change the units (mm x mm x mm).Author Response
The manuscript under the title: “Valorization of Linen Processing By-Products and Wastes for the Development of Injection-Molded Green Composite Pieces of Polylactide with Improved Performance” is appropriate for Sustainability journal. The authors present original research works, supported by experiment. The aim of article is precisely formulated. The research methodology is clearly described. The authors applied wide range of experimental research. The topic of the article is up-to-date. The organization of the article is appropriate. The abstract presents all required information and it is sufficiently informative. Overall, the paper is well prepared. Neverthless, some improvements could be applied:
Thank you for reviewing our manuscript.
Q1) line: 39-43 - please rewrite this sentence
A1) This sentence was rewritten. Please see lines 40-44.
Q2) line: 62 - both version are correct, but "the Ukraine" is most common,
A2) This indication was added. Please see lines 62-64.
Q3) line: 114, 148, 152 and 191 - please change the units (mm x mm x mm).
Units format was changed in the Experimental section.
Round 2
Reviewer 2 Report
The authors have addressed the raised issues. The reviewer has no further concern regarding the current work. It can now be accepted for publication.